

# Southern Ocean deep mixing band emerges from a competition between winter buoyancy loss and upper stratification strength

Romain Caneill[1], Fabien Roquet[1], and Jonas Nycander[2]

[1]Department of Marine Sciences, University of Gothenburg, Göteborg, Sweden
[2]Department of Meteorology, Stockholm University, Stockholm, Sweden

**Correspondence:** Romain Caneill (romain.caneill@gu.se)

**Abstract.** The Southern Ocean hosts a winter deep mixing band (DMB) near the Antarctic Circumpolar Current's (ACC) northern boundary, playing a pivotal role in Subantarctic Mode Water formation. Here, we investigate what controls the presence and geographical extent of the DMB. Using observational data, we construct seasonal climatologies of surface buoyancy fluxes, Ekman buoyancy transport, and upper stratification. The strength of the upper ocean stratification is determined using

the columnar buoyancy index, defined as the buoyancy input necessary to produce a 250 m deep mixed layer. It is found that the DMB lies precisely where the autumn – winter buoyancy loss exceeds the columnar buoyancy found in late summer. The buoyancy loss decreases towards the south, while in the north, the stratification is too strong to produce deep mixed layers. Although this threshold is also crossed in the Agulhas current and East Australian current regions, advection of buoyancy is able to stabilise the stratification. The Ekman buoyancy transport has a secondary impact on the DMB extent due to the com-

pensating effects of temperature and salinity transports on buoyancy. Changes in surface temperature drive spatial variations of the thermal expansion coefficient (TEC). These TEC variations are necessary to explain the limited meridional extent of the DMB. We demonstrate this by comparing buoyancy budgets derived using varying TEC values with those derived using a constant TEC value. Reduced TEC in colder waters leads to decreased winter buoyancy loss south of the DMB, yet substantial heat loss persists. Lower TEC values also weaken the effect of temperature stratification, partially compensating for the effect

of buoyancy loss damping. TEC modulation impacts both the DMB characteristics and its meridional extent.

## 1 Introduction

The Southern Ocean (SO) plays a crucial role in global climate dynamics (Rintoul, 2018). The Antarctic Circumpolar Current (ACC), a defining feature, is demarcated by key fronts (Fig. 1): the Northern Boundary (NB), the Subantarctic Front (SAF), the Polar Front (PF), the Southern Antarctic Circumpolar Current Front (SACCF), and the Southern Boundary (SB) (Orsi et al.,

1995). These fronts mark distinct regions of water masses with varying properties and stratification (Pauthenet et al., 2017). Temperature is the stratifying agent north of the SAF in the Atlantic and Indian sectors of the SO (Pollard et al., 2002), a regime known as the alpha ocean (Carmack, 2007). Between the SAF and the PF, both temperature and salinity increase stratification in the so-called polar transition zone (Caneill et al., 2022). Finally, salinity is the only stratifying agent in the beta ocean south of the PF.



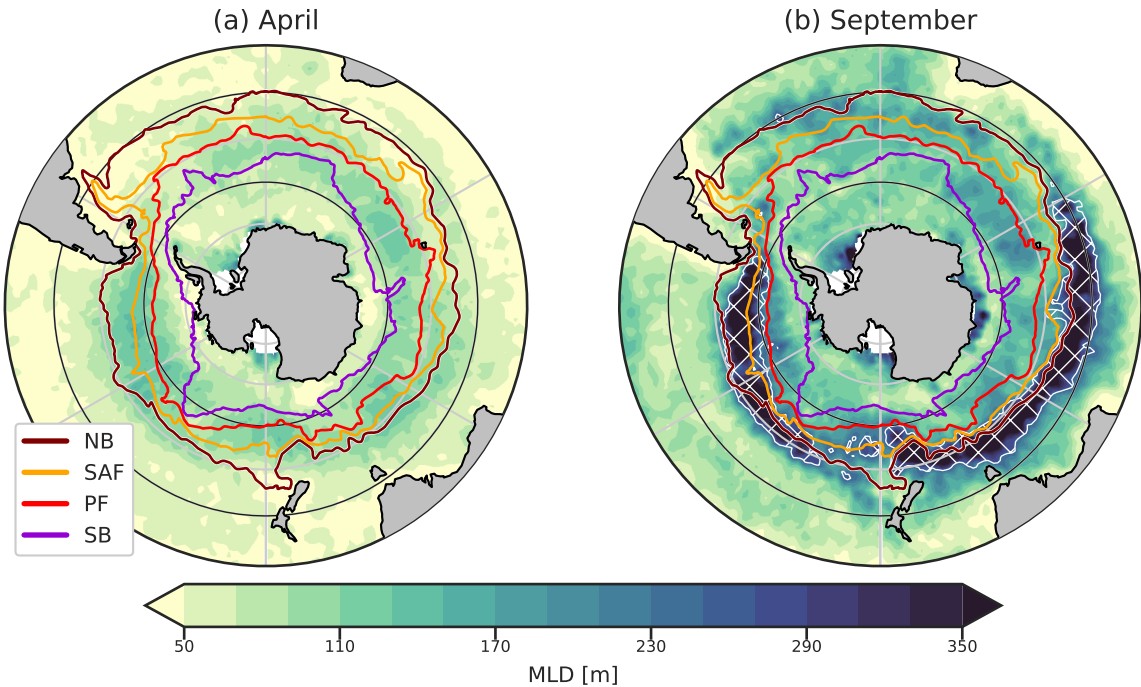

**Figure 1.** Mixed layer depth climatology in the SO with the major ACC fronts in (a) April and (b) September. From north to south, the NB (maroon colour), SAF (yellow), PF (red), and SB (purple) fronts from Park et al. (2019) are plotted. The MLD are from de Boyer Montégut (2023). Hatches and white contour represent the DMB. For these maps, and for the following maps of this paper, the northern boundary is at 30 °S. In latitude, grid lines are spaced every 10 degrees, with black lines at 40 °S and 60 °S.

A striking phenomenon known as the deep mixing band (DMB) emerges during winter in the SO (DuVivier et al., 2018). Situated mostly north of the SAF (Fig. 1 and Dong et al. (2008)), this narrow band of intense vertical mixing is of paramount importance in shaping the ocean's thermal and dynamic structure. The DMB is characterised by mixed layers (MLs) deeper than 250 m in winter. Intense buoyancy loss due to winter heat release deepens the ML, and the Ekman transport of cold water intensifies it (Naveira Garabato et al., 2009; Holte et al., 2012; Rintoul and England, 2002). This distinctive feature is found in

both the Indian and Pacific sectors of the SO and underpins the formation of the Subantarctic Mode Water (SAMW) (Belkin and Gordon, 1996; Speer et al., 2000; Hanawa and Talley, 2001; Klocker et al., 2023). The SAMW, a conduit for heat and carbon exchange, exerts significant influence on the global oceanic circulation and climate processes as it forms a major component of the upper limb of the global overturning circulation (Sloyan and Rintoul, 2001). By its capacity to take up anthropogenic $CO_2$ it is a key component of the Earth's climate (Sabine et al., 2004).

The DMB is a major site for the uptake of anthropogenic heat and carbon from the surface to the interior (Roemmich et al., 2015; Gruber et al., 2019), however the physics controlling both its location and latitudinal extent remains insufficiently un-





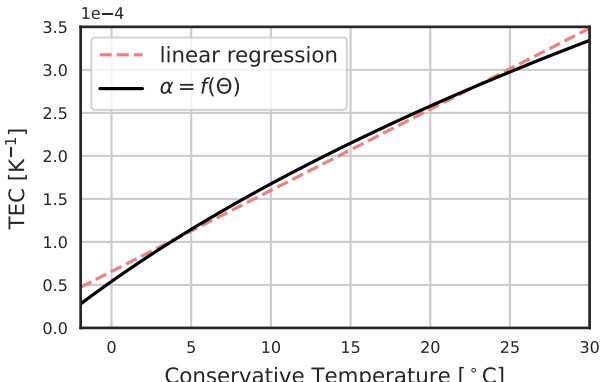

**Figure 2.** Thermal Expansion Coefficient function of Conservative Temperature. The pressure is $0 \, \mathrm{dbar}$ and the Absolute Salinity $35.5 \, \mathrm{g \, kg^{-1}}$. The graph stops at $-1.9 \, ^{\circ}\mathrm{C}$, the freezing point.

derstood (DuVivier et al., 2018; Fernández Castro et al., 2022). Holte et al. (2012) argued that the strength of the summer stratification determines the potential of the water column to form a deep ML in the following winter. The stratification in summer is decreased by enhanced vertical mixing (Sloyan et al., 2010), eddy-driven jet scale overturning circulation (Li and

Lee, 2017), and wind stress curl induced upwelling (Dong et al., 2008). Downstream of the Agulhas Retroflection and downstream of the Campbell Plateau, eddy diffusion of heat has a tendency to stabilise the mixed layer (Sallée et al., 2006, 2008), and the spring and summer heat gain contributes to increase stratification. The geostrophic flow is roughly oriented along the isotherms and does not induce any large horizontal heat fluxes in the SO, away from western boundary currents (Dong et al., 2007). In contrast, a subsurface salinity maximum advected from subtropical water is present in most of the SO in summer,

decreasing stratification below it (DuVivier et al., 2018). This salinity-driven weakening of stratification could be compensated by temperature stratification.

Another effect relates to the nonlinear nature of the equation of state for seawater, known to play a role for the formation of Antarctic Intermediate Water (Nycander et al., 2015). The density change induced by temperature is scaled by the thermal expansion coefficient (TEC, $\alpha$), defined as:

$$\alpha = -\left. \frac{1}{\rho} \frac{\partial \rho}{\partial \Theta} \right|_{S_{\mathrm{A}}, p} \qquad (1)$$

using Conservative Temperature $\Theta$ and Absolute Salinity $S_{\mathrm{A}}$ (IOC et al., 2015). The TEC also scales the contribution of heat fluxes to buoyancy fluxes. The TEC is a function of temperature and pressure and follows a quasi-linear relationship with temperature at the surface (Fig. 2). It is about 10 times smaller at $-1.8 \, ^{\circ}\mathrm{C}$ than at $30 \, ^{\circ}\mathrm{C}$. This effect makes the density of cold water less sensitive to temperature changes than that of warm water, enhancing the role of salinity and minimising the influence

of heat fluxes on buoyancy fluxes in polar regions (Bryan, 1986; Aagaard and Carmack, 1989; Roquet et al., 2015). Variations in the TEC value also imply that winter heat fluxes have less impact on density than summer heat fluxes. Schanze and Schmitt (2013) showed that variations in the TEC increase the global buoyancy flux by 35% due to seasonal effects. The variations in





the value of the TEC allow for a total non-zero buoyancy flux while having a zero net heat flux (Garrett et al., 1993; Zahariev and Garrett, 1997; Hieronymus and Nycander, 2013). The global stratification distribution is very sensitive to changes in the

TEC value (Roquet et al., 2015; Nycander et al., 2015). Its low value in the polar regions is the fundamental mechanism that allows the maintenance of a halocline under large cooling conditions and thus strongly promotes sea ice formation (Roquet et al., 2022).

Using numerical simulations of an idealised closed basin, Caneill et al. (2022) studied the impact of buoyancy fluxes in setting the position of the deep MLs, the equivalent of a DMB in their simulations. In their closed basin study, they determined

that the inversion of the sign of annual buoyancy fluxes primarily drives the poleward extent of the deep mixed layers. These buoyancy fluxes resulted from a competition between heat loss and freshwater gain. The inversion was, however, not driven by either an increase in freshwater fluxes or a decrease in heat loss. Rather, it was driven by the decrease in the TEC value in cold water, which strongly reduced the heat flux contribution to buoyancy fluxes towards the pole. The competition between heat and freshwater fluxes is thus unequal in polar regions. It is not clear whether a simple link between the DMB position

and buoyancy fluxes still holds in the SO and if the variations of the TEC are sufficient to constrain the poleward extent of the DMB. In fact, the position of the DMB is not correlated with a unique component of the surface forcing, which includes wind stress, wind stress curl, buoyancy flux, and mesoscale eddy activity (DuVivier et al., 2018).

Here, we aim to characterise the dominant control of the position of the DMB in the SO by the interplay of buoyancy fluxes and stratification. For this, we create climatologies of buoyancy fluxes and stratification and compare their respective

intensities. The effect of the Ekman transport is added as an extra term in the buoyancy fluxes. Measuring surface buoyancy fluxes presents challenges, entailing significant uncertainties within the SO (Cerovečki et al., 2011; Swart et al., 2019), so we focus on the climatological state of the ocean. Both seasonal and annual fluxes are taken into account, and the focus for stratification is late summer, after preconditioning has occurred, just before the ML starts to deepen. Additionally, we also aim at characterising the impact of the variations in the TEC value on the width of the DMB. For this, we compare the climatologies

computed with modified TEC values.

This paper is organised as follows: First, we present the data used in this study; we then describe the equations that govern buoyancy fluxes, the Ekman transport, and stratification intensity. We continue by describing the results and we end with a discussion and conclusions.

## 2   Data and methodology

Here, we describe how we constructed the climatologies for buoyancy fluxes and stratification (both its strength and thermohaline components). The general methodology consists in computing monthly components of the buoyancy fluxes for each year before averaging to produce monthly climatologies. We used the years 2005 to 2016 for all fluxes, as this is the record period of the ISCCP-FH MPF product (providing short and long wave heat fluxes). The stratification is computed using the Monthly Isopycnal & Mixed-layer Ocean Climatology (MIMOC) (Schmidtko et al., 2013).



### 2.1 Buoyancy fluxes

#### 2.1.1 Surface buoyancy fluxes

Surface buoyancy fluxes are defined from heat and freshwater fluxes (e.g. Gill and Adrian, 1982):

$$\mathcal{B}^{surf} = \underbrace{\frac{g\alpha}{\rho_0 C_p} Q_{tot}}_{\mathcal{B}^{surf}_\Theta} - \underbrace{\frac{g\beta S}{\rho_0}(E - P - R)}_{\mathcal{B}^{surf}_S} \tag{2}$$

where $g$ is the gravity acceleration, $C_p \simeq 3997\,\mathrm{J\,kg^{-1}\,K^{-1}}$ the heat capacity, $Q_{tot}$ the total heat flux in $\mathrm{W\,m^{-2}}$, $E$ the evaporation, $P$ the precipitation, $R$ the river runoff both in $\mathrm{kg\,m^{-2}\,s^{-1}}$, and $S$ the surface salinity. $\mathcal{B}^{surf}_\Theta$ is the heat contribution, $\mathcal{B}^{surf}_S$ the salt contribution, and $\mathcal{B}^{surf}$ the total surface buoyancy flux in $\mathrm{m^2\,s^{-3}}$.

$Q_{tot}$ is divided into 4 components: net long wave radiation ($Q_{LW}$), net short wave radiation ($Q_{SW}$), latent heat ($Q_{LH}$), and sensible heat ($Q_{SH}$). Monthly latent and sensible heat fluxes are taken from the monthly means Objectively Analyzed Air–Sea Fluxes 1 degree dataset (OAflux; Yu and Weller, 2007) (these fluxes are only provided for oceans free of ice) and ISCCP-FH MPF (monthly means) for the short and long waves (Schiffer and Rossow, 1983; Rossow and Schiffer, 1999). The TEC is computed using the SST provided by OAflux, and the surface salinity is taken from Estimating the Circulation and Climate of the Ocean (ECCO) Version 4, Release 4 (Forget et al., 2015). ECCO monthly outputs have been bilinearly interpolated from the tiles onto a 1-degree longitude–latitude grid using the SciPy Python library (Virtanen et al., 2020).

Due to the difficulties in computing the turbulent fluxes, the total heat budget is not closed. Following Schanze and Schmitt (2013), we remove the global average (time, longitude, and latitude) of the heat and freshwater fluxes to balance them. Before adjustment, the average of the heat fluxes is $24.49\,\mathrm{W\,m^{-2}}$, and the average integrated freshwater imbalance is $-0.2\,\mathrm{Sv}$. Compared to the heat and buoyancy fluxes derived from ECCO, it is found that the patterns of heat and buoyancy gain or loss are better represented after correction (Appendix A).

Freshwater fluxes combine OAflux for evaporation, Global Precipitation Climatology Project (GPCP) Version 2.3 for precipitation (Adler et al., 2018), and ECCO for river runoff.

Two seasonal components are necessary to complement the annual mean of the fluxes: buoyancy loss and buoyancy gain. We split the annual buoyancy flux into its negative and positive components: the "cooling season" (CS) and the "warming season" (WS). CS is defined by the months April to September (included) and WS by the months October to March (included). This gives $\mathcal{B}^{CS}$ and $\mathcal{B}^{WS}$. This division captures an overview of buoyancy loss and gain throughout the seasons.

Sea ice formation and melting are associated with freshwater fluxes at the surface of the ocean. In the Southern Ocean, sea ice mostly forms close to the Antarctic coast and melts more uniformly after being exported away by wind and oceanic currents (Holland and Kwok, 2012). In the sea ice covered area, winter heat fluxes are small, and brine rejection when sea-ice is forming is the main component of buoyancy fluxes (Klocker et al., 2023). In summer, buoyancy fluxes are positive as sea ice melts and the ocean warms up (Pellichero et al., 2018). In this study, we focus on the DMB located north of and within the ACC, which is ice-free all year. Therefore, we will not take sea ice formation or melting into account in our buoyancy flux calculations.



### 2.1.2 Ekman buoyancy fluxes

The Ekman transport advects heat and salt, which leads to lateral buoyancy fluxes. As is commonly done, we assume that the Ekman depth is contained within the ML (e.g. Qiu and Kelly, 1993; Sallée et al., 2006; Dong et al., 2007), so Ekman transport acts to modulate the buoyancy forcing seen at the bottom of the ML.

Associating the Ekman depth with the MLD comes from the fact that the turbulent viscosity should be large in the ML and small below. However, this is not obvious; studies do not always agree, depending on the region and method. Lenn and Chereskin (2009) found that viscosity decreased from approximately $0.1\,\mathrm{m^2\,s^{-1}}$ at $26\,\mathrm{m}$ depth to near zero at $90\,\mathrm{m}$ in the Drake passage using ADCP measurement of velocity. In contrast, Roach et al. (2015) found that a model with a vertically uniform viscosity between $0.05\,\mathrm{m^2\,s^{-1}}$ and $0.25\,\mathrm{m^2\,s^{-1}}$ represented the observed spirals well. Nevertheless, despite the lack

of observations of the Ekman depth in the SO, Dong et al. (2007) found that using the MLD as the Ekman depth was giving accurate results in the mixed layer heat budget.

To compute buoyancy fluxes, one needs to compute the Ekman horizontal transport (e.g. Qiu and Kelly, 1993; Yang, 2006):

$$-fV_{\mathrm{Ek}} = \frac{\tau^x}{\rho} \quad \text{and} \quad fU_{\mathrm{Ek}} = \frac{\tau^y}{\rho} \tag{3}$$

Using the heat and freshwater fluxes due to horizontal Ekman advection,

$$Q_{Ek} = \rho_0 C_p \boldsymbol{U_{Ek}} \cdot \boldsymbol{\nabla} \Theta \tag{4}$$

$$FW_{Ek} = \frac{\rho_0}{S} \boldsymbol{U_{Ek}} \cdot \boldsymbol{\nabla} S \tag{5}$$

the Ekman transport is derived from wind stress using (3), leading to:

$$\mathcal{B}_{\Theta}^{Ek} = -\frac{g\alpha}{\rho_0 f} \left( \tau^y \frac{\partial \Theta}{\partial x} - \tau^x \frac{\partial \Theta}{\partial y} \right) \tag{6}$$

$$\mathcal{B}_{S}^{Ek} = \frac{g\beta}{\rho_0 f} \left( \tau^y \frac{\partial S}{\partial x} - \tau^x \frac{\partial S}{\partial y} \right) \tag{7}$$

$\mathcal{B}^{Ek}$ are fluxes within the Ekman layer. Under the assumption that the Ekman layer is contained within the ML, $\mathcal{B}$, the sum of $\mathcal{B}^{Ek}$ and $\mathcal{B}^{surf}$, corresponds to the buoyancy flux taken at the base of the ML.

We computed daily averages of wind stress from the CMEMS "Global Ocean Wind L4 Reprocessed 6 Hourly Observations". To compute horizontal gradients of temperature and salinity, we used the ARMOR3D weekly product (Guinehut et al., 2012). We took the average of the three upper levels ($0\,\mathrm{m}$, $5\,\mathrm{m}$, and $10\,\mathrm{m}$) and then did a linear interpolation in time to get daily data.

## 2.2 Stratification from columnar buoyancy

We quantify the stratification using the columnar buoyancy (Lascaratos and Nittis, 1998; Herrmann et al., 2008):

$$CB(Z) = \int\limits_{Z}^{0} -zN^2(z)\mathrm{d}z \tag{8}$$



with the z-axis oriented upward. $N^2(z)$ is calculated using the locally referenced potential density. For shallow depths, an approximation using the potential density referenced at the surface is applicable. By integrating Eq. (8) with this approximation, a formula directly using potential density profiles is obtained:

$$CB(Z) \simeq \frac{g}{\rho_0} \int\limits_Z^0 [\rho_\theta(Z) - \rho_\theta(z)] \, \mathrm{d}z \tag{9}$$

The columnar buoyancy, positive under stable stratification, quantifies the vertically integrated amount of buoyancy that must be lost to create a ML of depth $Z$. We compute this index in April, which corresponds to the most stratified water column before it is eroded during the cooling season. Dividing $-CB$ by a time $\Delta t$ gives the buoyancy fluxes necessary to deepen the ML to $Z$ during $\Delta t$ (Faure and Kawai, 2015). Following DuVivier et al. (2018), we use the depth of 250 m as the deep mixed layer threshold:

$$B_{250} = \frac{-CB(-250)}{\Delta t} \tag{10}$$

We used $\Delta t = 6$ months to allow comparison with the mean buoyancy loss of the cooling season (Sect. 2.1.1).

The influence of temperature and salinity alone on stratification is done by separating their effects, as is similarly done by Sterl and De Jong (2022). $B_{250}$ can thus be split using the decomposition of the buoyancy frequency into the thermal and haline components:

$$B_{250} = \underbrace{\frac{g}{\Delta t} \int\limits_{-250}^0 \alpha(z) \frac{\partial \Theta}{\partial z} z \mathrm{d}z}_{B_{250}^\Theta} - \underbrace{\frac{g}{\Delta t} \int\limits_{-250}^0 \beta(z) \frac{\partial S}{\partial z} z \mathrm{d}z}_{B_{250}^S} \tag{11}$$

We expect $B_{250}^\Theta$ to be positive north of the PF and negative south of it, while $B_{250}^S$ should be negative north of the SAF and positive south of it.

The columnar buoyancy is computed using Monthly Isopycnal & Mixed-layer Ocean Climatology (MIMOC) in depth coordinates (Schmidtko et al., 2013).

## 2.3 Effect of the variations of the TEC

The TEC scales the effect of heat fluxes on buoyancy fluxes and determines the importance of temperature for stratification. To investigate how the TEC variations affect stratification and buoyancy fluxes, we computed them using $\alpha_0$, a constant TEC characteristic of the waters in the DMB. This replacement is made in Eqs. (2), (6), and (11). It will be explicitly mentioned when the constant TEC is used and the results are presented in Sect. 4.

To compute the value of $\alpha_0$, we used the average temperature of surface water located at $40\,°\mathrm{S}$ ($\Theta \simeq 14.7\,°\mathrm{C}$), a salinity of $35.5\,\mathrm{g\,kg^{-1}}$, and a pressure of $0\,\mathrm{dbar}$. The result is $\alpha_0 \simeq 2.1 \times 10^{-4}\,\mathrm{K^{-1}}$. The latitude $40\,°\mathrm{S}$ has been chosen as it roughly corresponds to the latitude located just north of the DMB.





## 3 Results

### 3.1 Annual buoyancy fluxes

Annual surface buoyancy fluxes induced by heat fluxes ($\mathcal{B}_\Theta^{surf}$) tend to be negative (cooling) north of $50\,°$S and positive south of it, but the pattern shows large regional disparities (Fig. 3 (a)). The DMB is associated with a net annual cooling; in the Indian and Pacific sectors, it is surrounded by regions of net annual warming. The heat loss is located north of the ACC in the Atlantic and Indian sectors, while the heat loss in the east Pacific sector is located within the ACC. Except for this large region of loss in the Pacific sector, even if the heat fluxes look patchy, heat is mostly gained within the ACC. This is due to large radiative heat fluxes compared to small sensible heat fluxes (Czaja and Marshall, 2015). Thus, heat fluxes are not zonally constant in the ACC (Song, 2020; Josey et al., 2023). Overall, in both the Indian and Pacific sectors of the SO, the DMB is surrounded by annual heat gain in the north and south. We note that the regions with the largest annual surface heat loss are colocated with the western boundary currents, due to the advection of warm water (Fig. 3 (a)).

In contrast to the patchiness and geographical variability of $\mathcal{B}_\Theta^{surf}$, the distribution of the haline component of the surface buoyancy fluxes, $\mathcal{B}_S^{surf}$, is much smoother. Precipitation in the south and evaporation in the subtropical gyres are responsible for the global pattern of freshwater fluxes (Fig. 3 (b)). $\mathcal{B}_S^{surf}$ is mainly positive within the DMB and positive everywhere south of it. This illustrates the fact that freshwater forcing depends little on ocean conditions, contrary to heat flux.

The sum of these two surface components, $\mathcal{B}^{surf}$, is similar to $\mathcal{B}_\Theta^{surf}$ north of the ACC (Fig. 3 (c)). Within and south of the ACC, it is positive everywhere except in the Pacific sector, where the DMB occurs: a narrow band of buoyancy loss is present with a large tongue of buoyancy gain north of it. Overall, a circumpolar band of buoyancy loss is present in the SO, surrounded by buoyancy gain, particularly in the Indian and Pacific sectors. Caneill et al. (2022) found that the poleward boundary of the deep MLs region was located at the inversion of the annual buoyancy fluxes in a coarse-resolution idealised closed basin study. The circulation in the SO differs from the one in a closed basin as considered in Caneill et al. (2022); the path of the ACC is more or less zonal, while the closed basin circulation has an important meridional upper ocean transport. Moreover, the ACC interacts with the bathymetry and presents a large eddy activity, two phenomena not really represented in the idealised basin of Caneill et al. (2022). Despite these differences, their conclusion that deep MLs (the DMB in the SO) are bounded poleward by the inversion of the annual buoyancy fluxes still holds in the SO. Additionally, the DMB is also bounded equatorward by tongues of annual buoyancy gain, so that the DMB coincides with a narrow band of annual buoyancy loss surrounded by buoyancy gain.

### 3.2 Effect of the Ekman transport

The Ekman transport brings cold water northward south of about $30\,°$S and thus creates a negative buoyancy flux at the base of the Ekman layer (Fig. 4 (a)). The largest negative values are found around the NB of the ACC in the Indian Ocean. The SAF has a large SST meridional gradient (Kostianoy et al., 2004), which enhances the meridional advection of cold water and thus creates a large negative buoyancy flux. The Ekman salt component presents large positive values close to the NB (Fig. 4 (b)). Summed together, the buoyancy fluxes induced by the Ekman transport are negative in the SO and positive in the southern part



**Figure 3.** Climatology of the annual components of the surface buoyancy fluxes. The heat component (a), haline component (b), and their sum (c) are plotted. On every plot, the thin black lines represent the 0 of the fluxes, and the maroon line is the northern boundary of the ACC, defined as the northernmost closed contour of Mean Dynamic Topography, as defined by Park et al. (2019). Hatches represent the DMB. The northern boundary is at $30\,°S$. In latitude, grid lines are spaced every 10 degrees, with black lines at $40\,°S$ and $60\,°S$.





of the subtropical gyres (Fig. 4 (c)). The negative temperature anomaly in the ocean induced by the Ekman transport creates a positive anomaly in surface heat fluxes. As a result, Ekman and atmospheric heat fluxes likely partially compensate for each 210 other. Such partial compensation can be seen when comparing Fig. 3 (a) and Fig. 4 (a) and (d), e.g., in the south-east Pacific Ocean, north of the ACC. When the Ekman contribution is included, almost all the SO south of $30\,°$S is losing buoyancy due to heat on the annual mean.

Regions within the ACC that gain heat from the atmosphere encounter compensation from Ekman transport and are thus also losing heat on the annual mean (Fig. 4 (d)). Although $\mathcal{B}_\Theta^{surf}$ is overall positive south of the NB of ACC, when the Ekman 215 heat component is included, $\mathcal{B}_\Theta^{tot}$ becomes negative almost everywhere in the SO (Fig. 4 (d)). The Ekman heat transport thus has a destabilising effect that counteracts the surface heat gain. The total salt component resembles its atmospheric part while being slightly modulated by Ekman transport. $\mathcal{B}$ is negative between the ACC and $25\,°$S, and south of the DMB, it alternates between regions of positive and negative values (Fig. 4 (f) and Fig. 5 (b)). It is continuously negative within the northern part of the ACC in the Pacific sector.

In short, the Ekman transport generates important fluxes of both temperature and salinity but the net effect on the buoyancy at the bottom on the ML remains limited overall, with a few regional exceptions such as around the Kerguelen Plateau.





**Figure 4.** Climatology of the annual components of the Ekman induced buoyancy fluxes, and the sum with the surface fluxes. The subplots are organized as follows: the columns show the Ekman fluxes and the sum of Ekman and surface. The rows show the thermal component, haline component, and their sum.



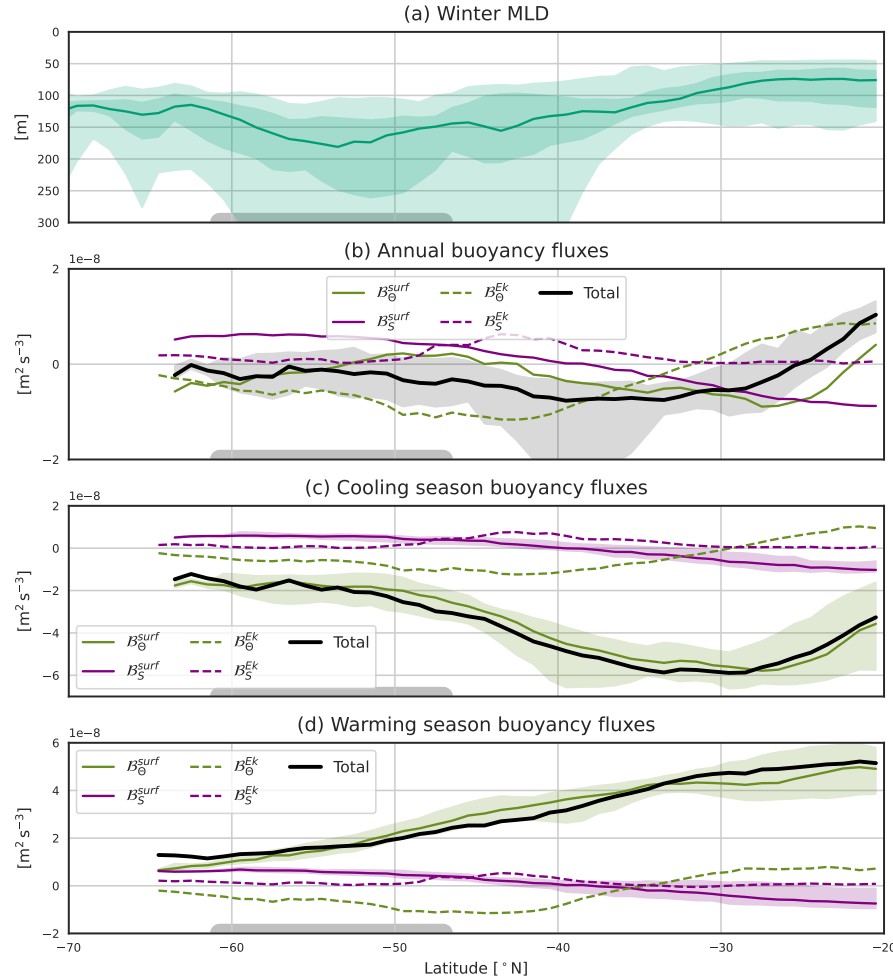

**Figure 5.** (a) Winter MLD. The shading represents respectively the 5th – 95th, and 25th – 75th percentiles, and the solid line is the zonal median. Mixed layers deeper than $250\,\mathrm{m}$ are found between $40\,°\mathrm{S}$ and $60\,°\mathrm{S}$ depending on the SO sector. (b) annual mean of the buoyancy flux components, (c) cooling season means, and (d) warming season means. The black line is $\mathcal{B}$ the sum of surface and Ekman fluxes. The vertical grid spacing is constant between panels. The black, green, and purple shading represents the region between the 25th and 75th percentiles for the total, heat surface, and salt surface components. The majority of the zonal differences arise from the surface heat component. The gray box in the bottom represents a median position of the ACC.





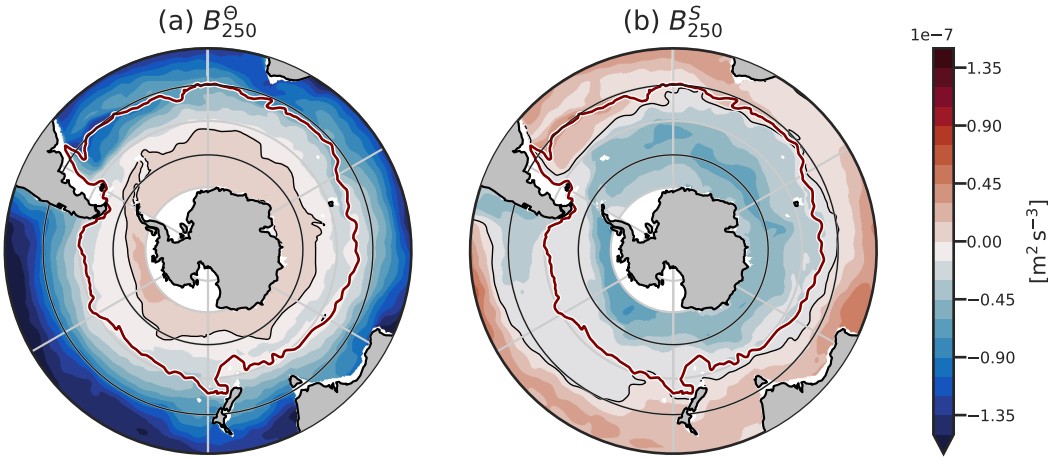

**Figure 6.** Climatology of (a) the thermal and (b) haline components of $B_{250}$.

### 3.3 Upper ocean stratification and seasonal cycle of buoyancy fluxes

The strength of the stratification is studied using the columnar buoyancy $B_{250}$ (computed in April, before the cooling season), i.e. the equivalent buoyancy flux necessary to erode the summer stratification and form a 250 m deep mixed layer. The subtrop-

ical and subpolar SO are thermally stratified ($B_{250}^{\Theta} < 0$), while in the polar region, the temperature component of stratification is negative (Fig. 6 (a)). The haline stratification follows an opposite pattern, with the salinity component of stratification decreasing stratification in the subtropical regions ($B_{250}^{S} > 0$) and increasing it in the polar regions (Fig. 6 (b)). We note that the transition from salinity destabilisation to stabilisation is located northward of the temperature inversion. The exact location of these stratification changes depends on the reference depth taken (250 m here), but the global pattern is a robust feature of the

stratification from subtropical to polar environments (Carmack, 2007; Roquet et al., 2022).

In the DMB, as well as everywhere except close to Antarctica, the temperature stratifies on average in the upper 250 m in summer (Fig. 6 (a)). The role of salinity is to stabilise the DMB in the Pacific Ocean; in the Indian Ocean, the DMB is located at the boundary between the salinity-stabilising and destabilising regimes. The presence of a salinity maximum advected from Agulhas water around 150 m deep in the region north of the ACC helps to decrease the summer stratification in the Indian

Ocean (Wang et al., 2014; Small et al., 2021; Fernández Castro et al., 2022).

The stratification is the weakest in the southern part of the SO (Fig. 7 (a), Fig. 8, and Fig. 9 (a)). The strongest stratifications are found in the subtropics (dark blue colours representative of the most negative value of $B_{250}$, Fig. 7 (a)). At 20 °S, it reaches a value more than five times larger than in the polar region. A band of stratification minimums exists around the DMB. Part of





**Figure 7.** Climatology of (a) the CS buoyancy fluxes, (b) the intensity of late summer stratification characterized by $B_{250}$, and (c) the difference $B_{250} - \mathcal{B}^{CS}$. The hatches surrounded by the black contour represent the DMB. The three black lines represent the transects plotted in Fig. 8.

the existence of this minimum can be attributed to the fact that $B_{250}$ includes the effect of the seasonal cycle of the stratification.

Outside the DMB, $B_{250}$ captures more permanent stratification than within the DMB. Weak stratification is also found outside



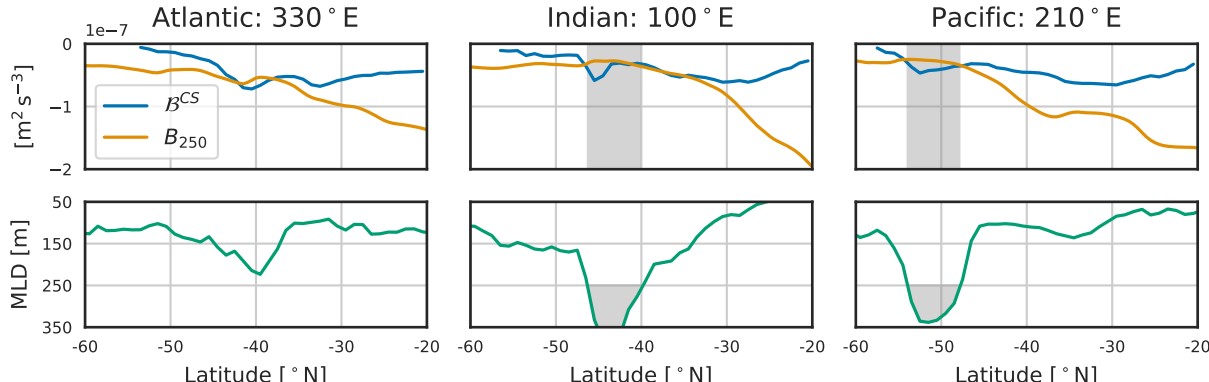

**Figure 8.** $B_{250}$ (orange curves) and $\mathcal{B}^{CS}$ (blue curves) (upper row), and observed winter MLD (green curve in lower row), for 3 different transects in the Atlantic, Indian, and Pacific sectors of the SO. The gray box represent the DMB.

the DMB, and the stratification minimum is not very pronounced. The general trend is that the stratification decreases from the subtropics to the polar region.

The annual mean of $\mathcal{B}^{surf}_{\Theta}$ hides a large seasonal cycle with a large winter buoyancy loss and a large summer buoyancy gain (Fig. B1 in Appendix B). In contrast, the other components have a very small seasonal cycle compared to $\mathcal{B}^{surf}_{\Theta}$ (Fig. 5 (c)

and (d)). To go beyond annual buoyancy fluxes, one can compare the seasonal fluxes with the position of the DMB (Fig. 5 (a)). One striking point is that the DMB (with MLDs deeper than $250\,\mathrm{m}$) is not located in regions of the largest annual or winter buoyancy loss but south of them. The largest CS buoyancy loss of about $-6 \times 10^{-8}\,\mathrm{m^2\,s^{-3}}$ is located around $30\,°\mathrm{S}$, where the winter MLD barely exceeds $100\,\mathrm{m}$. The winter buoyancy loss is three times lower at $50\,°\mathrm{S}$, a region where deep MLs are formed. Therefore, a large winter buoyancy loss itself is not sufficient to produce deep winter ML. This is because the poleward

advection of stratified water from the subtropics limits the ML deepening. Therefore, stratification strength also needs to be taken into account when studying deep ML formation.

Regions where $\mathcal{B}^{CS}$ is more negative than $B_{250}$ have a large probability of forming a ML deeper than $250\,\mathrm{m}$, unless another process not taken into account makes the water column more stable. The CS buoyancy loss driven by heat fluxes is the most intense around $30\,°\mathrm{S}$ ((Fig. 7 (b)), a region where the gain in summer buoyancy is also very large. Apart from the summer

buoyancy gain, this is also a region where subtropical stratified water is advected. South of it, the winter buoyancy loss becomes smaller, so it has less potential to form a deep mixed layer. The general pattern is that the buoyancy loss becomes smaller from $30\,°\mathrm{S}$ poleward.

Both the buoyancy loss and the stratification decrease poleward. To assert how the balance between these two opposite effects shapes the DMB, we compare the position of the DMB with the residual $B_{250} - \mathcal{B}^{CS}$ (Fig. 7 (c)). MLD larger than

$250\,\mathrm{m}$ are only found where $B_{250} > \mathcal{B}^{CS}$, but not vice versa. The largest mismatch is located south of Africa, in the Agulhas Retroflection region. Our analysis does not include the direct effect of advection by the geostrophic flow. However, our results strongly point to its restoring effect on the stratification in the regions of large western boundary currents such as the Agulhas



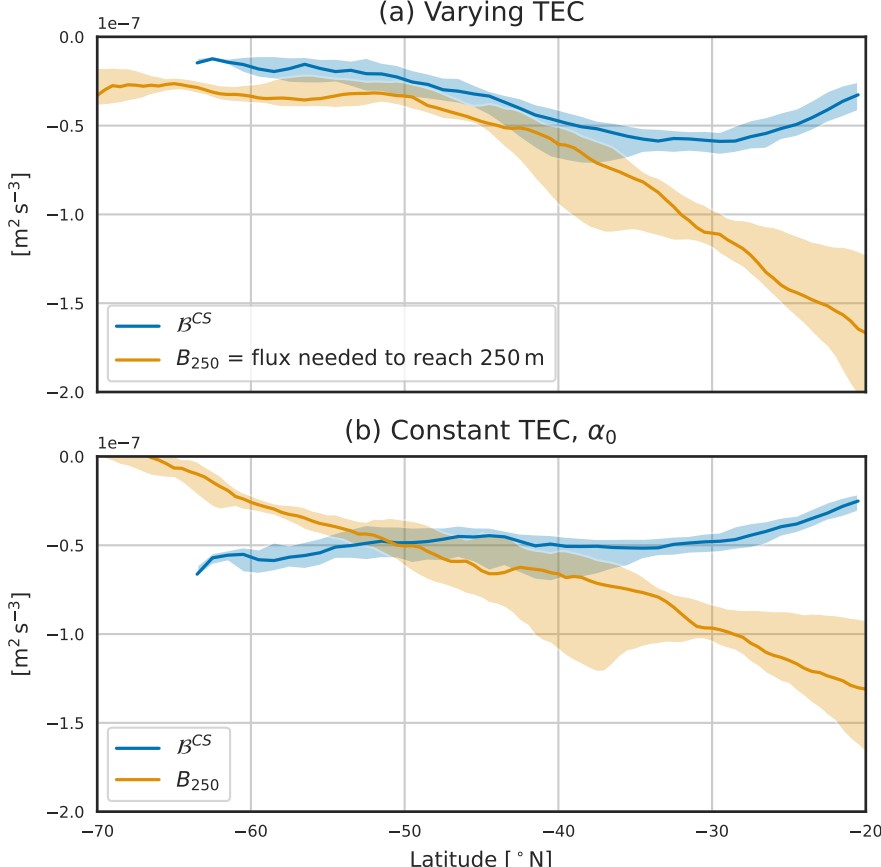

**Figure 9.** CS buoyancy fluxes (blue) and $B_{250}$, the buoyancy fluxes needed to produce a 250 m deep mixed layer (orange). (a) is computed using the realistic varying TEC and (b) is computed with $\alpha_0$. Shaded areas correspond to the 1st and 3rd quartiles, and the solid lines are the zonal medians. The blue line corresponds to the black line of Fig. 5 (c). This plot extends to 20 °S to highlight the increase of stratification towards the tropics, and the maximum buoyancy loss located around 30 °S.

Retroflection or the East Australian current. Overall, the simple balance between $B_{250}$ and $\mathcal{B}^{CS}$ as main drivers of deep ML formation does not hold at the western boundary currents but otherwise predicts with good accuracy the location of the DMB.

The zonal sections and the zonal median show the competition between the winter buoyancy loss and the existing summer stratification (Fig. 8 and Fig. 9 (a)). Sections in the middle of the Atlantic, Indian, and Pacific sectors of the SO are plotted. In the transect of the Atlantic sector, the MLD does not reach 250 m, but the location where buoyancy fluxes are more negative than stratification still corresponds to the position of the deepest MLD of the transect. It becomes clear that only in a few locations does the winter buoyancy loss become sufficient to erode the stratification.



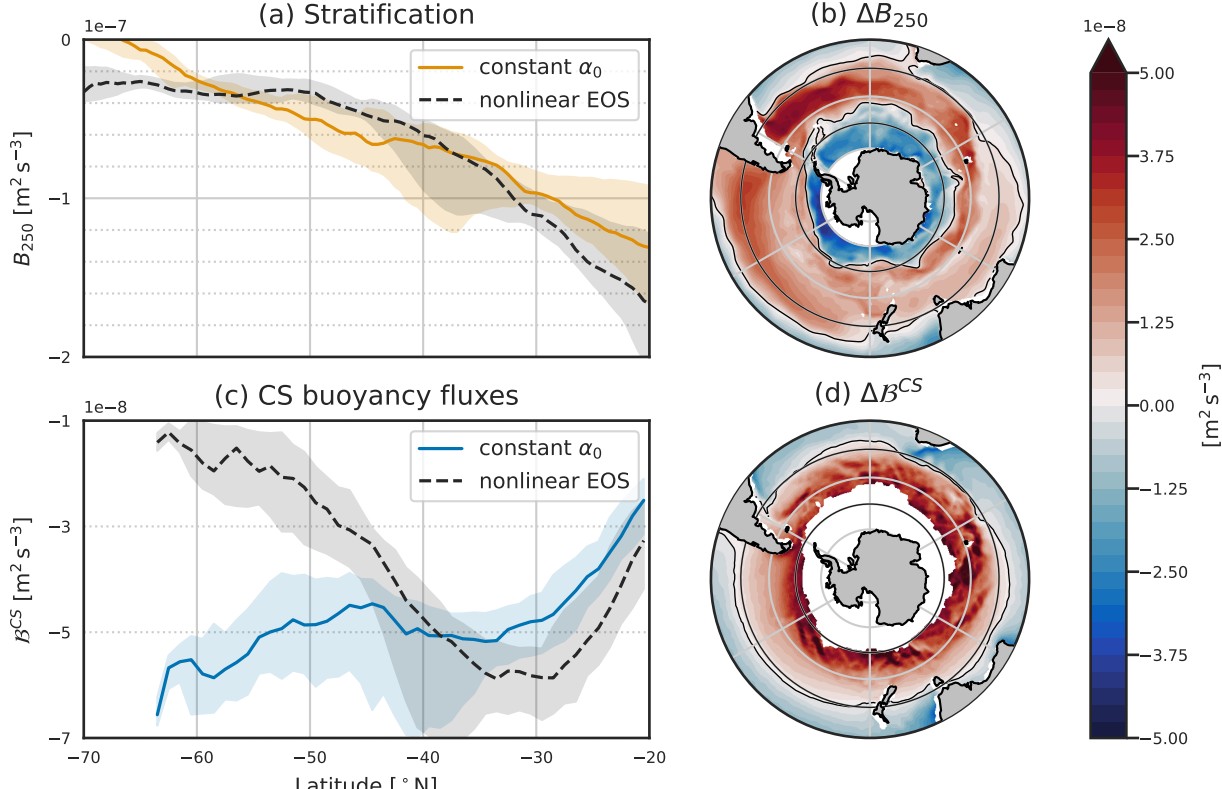

**Figure 10.** Stratification and buoyancy fluxes and computed using the nonlinear EOS or a constant $\alpha_0$. In panels (a) and (c), black dashed lines are for the nonlinear EOS, and blue or orange continuous lines are with constant $\alpha_0$. The spacing between the horizontal dotted gray line is the same in these two panels and equal to $2 \times 10^{-8} \, \mathrm{m^2 \, s^{-3}}$ The light shadings correspond to the 25th and 75th percentiles. Panels (b) and (d) are the difference between using the varying TEC and the constant $\alpha_0$, for $B_{250}$ and $\mathcal{B}^{CS}$ respectively.

Despite a weak stratification south of $50 \, ^\circ \mathrm{S}$ in the Atlantic and Indian sectors, no deep ML is observed as the winter buoyancy loss is smaller there. As seen previously, the heat component of the buoyancy fluxes is scaled by the TEC, itself a function of temperature. We will now explore the importance of this mechanism.

## 4 Effect of the local value of the TEC

To understand how variations in the TEC value affect the SO stratification, we computed the buoyancy fluxes and the columnar
buoyancy using a constant value $\alpha_0$, representative of $40 \, ^\circ \mathrm{S}$. We use the residual of buoyancy loss and stratification to predict the geographical extent that the DMB would have in this case.

Heat fluxes are negative in winter, so south of $40 \, ^\circ \mathrm{S}$, where $\alpha_0$ is larger than the real value of the TEC, the buoyancy fluxes become more negative (blue line in Fig. 9 (b) and Fig. 10 (b) and (c)). On the contrary, the buoyancy fluxes become smaller







**Figure 11.** Same as Fig. 7 but using $\alpha_0$ in computations. Hence, (c) represents where the ocean could form the DMB if the TEC was constant. The hatches represent the observed DMB. The region with sea-ice around Antarctica is masked in white.





north of $40\,°$S. This implies that the poleward extent of the DMB is primarily determined by the decrease in TEC value, which
significantly damps heat fluxes impact on buoyancy fluxes. It is not constrained by a decrease in winter heat loss to the south.

The TEC modulates the effect of temperature on stratification. North of $40\,°$S, the temperature stratifies and the TEC value
is larger than $\alpha_0$, so using $\alpha_0$ implies a weaker stratification (orange line in Fig. 9 (b) and Fig. 10 (a)). $\Delta B_250$, the difference
between the columnar buoyancy computed with the varying TEC and the constant TEC is thus negative (Fig. 10 (b)). In contrast,
between $40\,°$S and the PF, where temperature still decreases with depth and the TEC value is smaller than $\alpha_0$, stratification
is strengthened. South of the temperature inversion, where colder water lies above warmer water, an increased TEC value
decreases or even destabilises the stratification. Such regions are found where $\Delta B_{250} < 0$, close to Antarctica, as seen in
Fig. 10. The effect of the TEC on stratification is thus not constant over all the SO, but it is its small value that allows the
typical temperature inversion of polar regions while having a stable water column.

Where $\mathcal{B}^{CS}$ is more negative than $B_{250}$, deep MLs would be produced (Fig. 9 (b) and Fig. 11). The global pattern of
stratification strength changes slightly with the constant TEC (Fig. 11 (a)). The stratification is weakening continuously from
the subtropics towards Antarctica, until it eventually becomes negative and thus unstable. The winter buoyancy loss is intense
everywhere within the SO and does not decrease poleward (Fig. 11 (b)).

In the Indian Ocean, a second region where $B_{250} - \mathcal{B}^{CS} > 0$ appears south of the DMB (the red region close to the sea ice
edge). North of this second band, around $50\,°$S, even if $\mathcal{B}^{CS}$ becomes very negative due to the large value of $\alpha_0$, the stratification
is also slightly stronger, which counterbalances for the larger buoyancy loss. In the Pacific Ocean, where stratification is smaller,
the predicted DMB extends all the way to the sea ice edge. As the reference value for the TEC is $40\,°$S, the northern boundary
of the DMB does not change much.

The overall effect of using the constant TEC is that the southern boundary of the predicted DMB is shifted southward, up to
the edge of the sea ice. The DMB formed in an ocean with a constant $\alpha_0$ would overlap with the observed DMB (Fig. 11 (c)).
In the Indian Ocean, two DMBs are formed: one with the same position as the observed one, and one within the ACC, close
to the actual sea-ice edge. In the Pacific Ocean, these two DMBs are merged into a wide one. It is likely that such deep MLs
would be formed, as mean advection is more or less zonal and does not bring highly stratified waters within the ACC. The
narrowness of the DMB thus arises from the decrease of the TEC, that itself induces a decrease in winter buoyancy loss.





## 5   Discussion and conclusions

In this study, our goal was to determine if the position of the Southern Ocean DMB and its narrowness can be explained only by the balance between the buoyancy loss of the cooling season and the stratification intensity. We find that this balance is sufficient to predict the DMB away from western boundary currents. The narrowness of the DMB emerges because, south of it, despite a weakly stratified ocean, the buoyancy fluxes are small. This small buoyancy loss results from the decrease of the TEC in cold water and not from a decrease in winter heat loss towards the pole.

A perfect prediction of the DMB spatial extent is not realistic, as other processes such as wind-induced turbulence contribute to deepening the ML (Holte et al., 2012). Our study, however, highlights the first-order role of the competition between winter buoyancy loss and existing stratification in forming the DMB. North of $40\,°$S, despite the largest CS buoyancy loss, no deep ML is produced because the stratification of the water column induced by temperature is also very strong. In contrast, within the ACC (except in the East Pacific sector), the water column is only weakly stratified, but the CS buoyancy loss is small. Hence,

it is only in a narrow band that both winter buoyancy loss is large enough and stratification is weak enough to significantly deepen the ML.

The buoyancy fluxes follow a strong seasonal cycle, mainly driven by surface heat fluxes. South of $30\,°$S, the Ekman transport advects cold water northward, which produces a negative heat flux. It is likely that this flux itself induces a positive heat flux from the atmosphere on an annual scale. This could explain why the negative Ekman heat flux can counterbalance the surface

heat gain (Tamsitt et al., 2016). Ekman transport must therefore be taken into account when computing buoyancy fluxes in the mixed layer. Because of the variability of the wind stress, it drives a large part of the inter-annual variability of the SAMW temperature and salinity (Rintoul and England, 2002).

We have included the effect of the Ekman transport in the buoyancy fluxes. If the Ekman depth is below the MLD, a part of the buoyancy flux that we estimate will be brought below the ML. However, previous studies have found only minor

differences in the ML heat budget if taking the MLD as Ekman depth, or 1.5 times the MLD (Dong et al., 2007). Furthermore, the ageostrophic velocities within the Ekman layer decrease with depth, so most of the transport occurs closer to the surface than to the bottom of the Ekman layer. Thus, if the Ekman depth is slightly larger than the MLD, the transport within the ML will only be slightly overestimated.

Even if the annual buoyancy fluxes are a small residual compared to the seasonal fluxes, they will impact the buoyancy

budget of the water column along its path. The region within the ACC in the Pacific Ocean tends to lose buoyancy on an annual scale (Fig. 3 (f)) and stratification is smallest in the East Pacific sector (Fig. 7 (a)). This decrease in stratification could be partially attributed to the annual loss of buoyancy. In the Southeast Pacific sector, the low stratification of the upper ocean during summer is also linked to enhanced vertical diffusivity in the upper ocean (Sloyan et al., 2010).

The decrease of the TEC in cold water damps the effect of the heat component of the buoyancy fluxes. This also decreases

the contribution of temperature to stratification. Using a constant TEC for the calculation of buoyancy fluxes, we have shown that the region of large buoyancy loss would be located further poleward, while the stratification induced by temperature would increase if the TEC was constant. As a result of these two opposite effects, the DMB's southern boundary would be located





further poleward. This theoretical result has previously been observed in idealised model runs with various values for the polar TEC (Caneill et al., 2022). In line with Roquet et al. (2022), we find that by restraining the deepening of the ML, the variations

of the TEC strongly limit the rates of exchange between the surface and the ocean interior. South of the ACC, the impact of temperature is to decrease stratification beneath the winter mixed layer (Pollard et al., 2002). A higher TEC would amplify this effect, resulting in reduced stratification. Furthermore, a higher value of the TEC would enhance the impact of heat loss on buoyancy loss. Thus, climate change and an increase in the (sub)polar ocean temperature may lead to a further southward extent of the DMB.

A direct implication of our study is that one should not convert buoyancy and freshwater fluxes into heat fluxes. It is important to retain the scaling effect of the TEC on the heat component of the buoyancy fluxes. Taking the annual heat flux and converting it to buoyancy flux using the mean TEC or taking the mean buoyancy flux is not equivalent, as described by Schanze and Schmitt (2013). Thus, when studying stratification, heat and freshwater fluxes need to be converted into buoyancy or density fluxes to accurately quantify their effects on density.

In summary, this study provides a global view of the formation of the deep mixing band in the Southern Ocean. The balance between 1) stratification at the end of summer and 2) buoyancy loss during the cooling season is sufficient to explain the formation of the DMB, its position, and its width. The reduction of the thermal expansion coefficient value in cold water limits the influence of heat fluxes on buoyancy fluxes. Consequently, despite the large loss of heat in winter, stratification cannot be significantly eroded south of the actual DMB. The small stratification of the Southeast Pacific sector of the ACC is easily

eroded, and despite the small winter buoyancy loss in this region, deep MLs are formed in winter.

*Code and data availability.* ECCO has been downloaded from https://ecco.jpl.nasa.gov/drive (ECCO Consortium et al., 15-03-2022, 2021).

OAflux has be downloaded from ftp://ftp.whoi.edu/pub/science/oaflux/data_v3/monthly/ (accessed on 2022-02-24)

ISCCP-FH MPF has been downloaded from https://isccp.giss.nasa.gov/pub/flux-fh/tar-nc4_MPF/ (accessed on 2022-03-01).

GPCP has been downloaded from https://www.ncei.noaa.gov/data/global-precipitation-climatology-project-gpcp-monthly/access (accessed

on 2022-03-01).

The "Global Ocean Wind L492 Reprocessed 6 hourly Observations" has been replaced on November 2022 by the Global Ocean Hourly Reprocessed Sea Surface Wind and Stress from Scatterometer and Model, E.U Copernicus Marine Service Information (CMEMS), Marine Data Store (MDS), DOI: 10.48670/moi-00185. The data have been downloaded from https://data-cersat.ifremer.fr/data/ocean-wind/mwf/mwf-blended/reprocessing/v6/ (accessed on 2023-10-02).

ARMOR3D with DOI: 10.48670/moi-00052 (accessed on 2022-06-23) has been downloaded from

ftp://nrt.cmems-du.eu/Core/MULTIOBS_GLO_PHY_TSUV_3D_MYNRT_015_012/dataset-armor-3d-rep-weekly

MIMOC has been downloaded from http://www.pmel.noaa.gov/mimoc (accessed on 2021-12-09).

The MLD from de Boyer Montégut (2023) based on the work of de Boyer Montégut (2004) has been downloaded from https://www.seanoe.org/data/00806/91774/data/103667.tar (accessed on 2023-08-01).

The climatologies produced in this paper will be shared on Zenodo after the review process. During the review process, they are available upon request to the contact author. The scripts used to make the figures will be shared via Github and a Zenodo DOI after the review process.



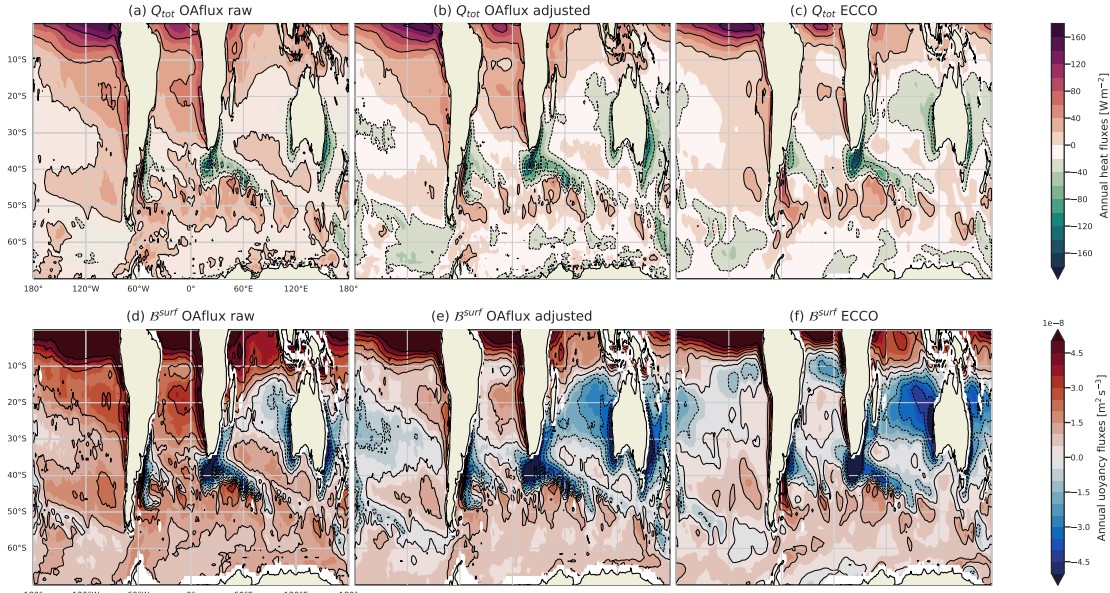

**Figure A1.** Heat fluxes (first row) and surface buoyancy fluxes (second row) are presented. The first column is the climatology annual mean without any correction, the second column is with our adjustment, and the 3rd column has been computed with the fluxes from ECCO.

During the review process, source code is available upon request to the contact author. All the analyses and figures are reproducible within a few steps.

## Appendix A:  Correction of heat fluxes

This appendix assesses the validity of the correction on heat fluxes we have applied. Due to the heat imbalance of OAflux, without correction the ocean gains too much heat compared to ECCO (Fig. A1). This is particularly visible in the SO: without correction, almost all the SO gains heat on average, whereas ECCO and our adjusted product show more patchiness and variability. This excess of heat gain is also visible in the buoyancy fluxes (Fig. A1 second row). For example, without the heat correction all Pacific sector of SO gains heat, while in ECCO and after correction a small band of buoyancy loss is present.

The comparison of the adjusted product with ECCO allows us to have confidence that our computations represent well the climatological state of the SO.





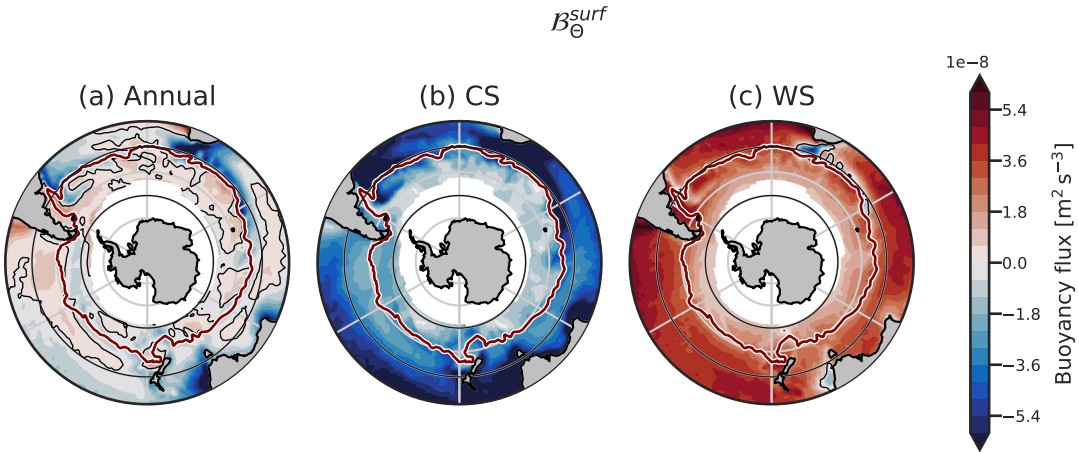

**Figure B1.** Climatology of the annual, Cooling Season, and Warming Season surface heat component of the buoyancy flux.

## Appendix B:  Extra maps of buoyancy fluxes



*Author contributions.* All authors contributed to the study conception. RC conducted analyses. RC prepared the manuscript with contributions from all co-authors.

*Competing interests.* The authors declare that they have no conflict of interest.

*Acknowledgements.* The analyses that lead to this publication are based on snakemake (Mölder et al., 2021), xarray (Hoyer and Hamman, 2017; Hoyer et al., 2023), xgcm (Abernathey et al., 2022), cf-xarray (Cherian et al., 2023), and thermodynamic computations were made using gsw-xarray, the xarray wrapper around GSW-Python (McDougall and Barker, 2011; Caneill and Barna, 2023).

This publication is based upon the WHOI OAFlux datasets supported by the NOAA's Global Ocean Monitoring and Observing (GOMO)
Program and NASA's Making Earth System Data Records for Use in Research Environments (MEaSUREs) Program.



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
