# Peer review of "Southern Ocean deep mixing band emerges from a competition between winter buoyancy loss and upper stratification strength"

_EGUsphere, 2023_

## Referee Comment (RC1)

**Review: "Southern Ocean deep mixing band emerges from a competition between winter buoyancy loss and upper stratification strength"**

Authors: Romain Caneill, Fabien Roquet, and Jonas Nycander

**Summary**

This study, using the observational monthly data and ECCO monthly outputs, explores what controls the position and narrowness of the Southern Ocean deep mixing band (DMB). The authors investigate two critical factors: the wintertime buoyancy loss and the stratification intensity. The net balance between these two factors is found to be sufficient to predict the DMB formation far from western boundary currents. Ekman buoyancy transport plays a secondary role, which can counterbalance the effect of surface buoyancy fluxes. Finally, the authors find that the spatial variations of thermal expansion coefficient (TEC) are necessary to explain the limited meridional extent of the DMB across the Southern Ocean.

The authors present some interesting results, highlighting the dominant role of surface buoyancy fluxes and the strength of the upper ocean stratification in the DMB formation. The paper is clear, well-written, and worthy of publication. However, I have some concerns on the paper's conclusions and methodology, which should be further addressed.

**Major comments**

(1) **Lines 305–307**: "*In this study, our goal was to determine if the position of the Southern Ocean DMB and its narrowness can be explained **only** by the balance between the buoyancy loss of the cooling season and the stratification intensity. We find that this balance is **sufficient** to predict the DMB away from western boundary currents.*"

(a) What is the defination of "***narrowness***" of DMB here? What is the horizontal resolution of observational data and ECCO outputs used in this study? Compared with the MLD simulated from an eddying (0.1°) ocean model (e.g., Fig. 2 in Li & Lee, 2017), the meridional extent of wintertime MLD (Fig. 1b) looks still quite broad to me. Note that the MLD from de Boyer Montégut et al. (2023) is at 1 °× 1 °spatial resolution. More discussion/clarification is needed here.

(b) The authors also need to clarify that "*this balance is **sufficient** . . .*" on the timescale of annual mean or 6-month-mean, which is the time period focused in this study. In fact, I am concerned on comparing the contributions of surface buoyancy fluxes and Ekman buoyancy transport on the timescale of 6-month or longer, as they both can play the dominate role of preconditioning in the deep MLD formation but during different periods. For example, the *May* net air-sea heat flux and *June* Ekman heat advection are both critical in the *August/September* MLD formation (Li & England, 2020).

(c) The first-order role of surface buoyancy fluxes is mostly considered on a large-scale or a zonal average (Fig. 5). In southeast Pacific, Ekman buoyancy transport actually dominates the deep MLD formation and Subantarctic Mode Water (SAMW) formation (Cerovečki et al., 2019; Li & England, 2020). More discussion is needed here.

(d) In the MLD budget, there are some other terms, such as vertical Ekman pumping and vertical mixing, that could be potentially important. More discussion is needed here.

(2) **Section 2.2**: I do not follow why the authors use **a deep mixed layer threshold of 250 m** in the estimation of stratification intensity. The wintertime MLD, that forms north of the Subantarctic Front (SAF), can be much deep over 500 m. The summertime MLD at southern high-latitudes can be less than 100 m. Thus, it is unjustified to apply this threshold across the entire Southern Ocean. I suggest to use the actual MLD in the calculations (Eq. 11). Then, this equation can be written as follows:

$$B_{\text{MLD}} = \frac{g}{\Delta t}(\int_{-\text{MLD}}^{0} \alpha(z)\frac{\partial \theta}{\partial z}zdz - \int_{-\text{MLD}}^{0} \beta(z)\frac{\partial S}{\partial z}zdz)$$

**Minor comments**

– Figure 8: The authors examine three different transects in the Atlantic, Indian, and Pacific sectors of the Southern Ocean. However, these transects cover a large domain, in which many different processes may mix together. I recommend changing each domain to that more localized in the deep MLD formation region. For example, the SAMW formation regions analyzed in Li et al. (2021) and Cerovečki et al. (2013).

– Equation (3): Define the $\tau^x$ and $\tau^y$.

- Equation (4)–(7): Define the $\theta$ and $S$.

- Equation (8): Define the $Z$ and $z$ right after this equation.

- Figure 6: Add "annual" to the figure caption.

- Line 331: Change Fig. 3 (f) to Fig. 4 (f).

- There are too many acronyms in the paper, and I suggest to reduce the use of them if possible. For example, I may suggest to spell out the "Southern Ocean (SO)", "cooling season (CS)", "warming season (WS)", etc.

**References**

Cerovečki, I., Meijers, A. J. S., Mazloff, M. R., Gille, S. T., Tamsitt, V. M., & Holland, P. R. (2019). The Effects of Enhanced Sea Ice Export from the Ross Sea on Recent Cooling and Freshening of the Southeast Pacific. *Journal of Climate*, *32*(7), 2013–2035.

Cerovečki, I., Talley, L. D., Mazloff, M. R., & Maze, G. (2013). Subantarctic Mode Water Formation, Destruction, and Export in the Eddy-Permitting Southern Ocean State Estimate. *Journal of Physical Oceanography*, *43*(7), 1485–1511.

Li, Q., & England, M. H. (2020). Tropical Indo-Pacific Teleconnections to Southern Ocean Mixed Layer Variability. *Geophysical Research Letters*, *47*(15), e2020GL088466.

Li, Q., & Lee, S. (2017). A Mechanism of Mixed Layer Formation in the Indo–Western Pacific Southern Ocean: Preconditioning by an Eddy-Driven Jet-Scale Overturning Circulation. *Journal of Physical Oceanography*, *47*(11), 2755–2772.

Li, Z., England, M. H., Groeskamp, S., Cerovečki, I., & Luo, Y. (2021). The Origin and Fate of Subantarctic Mode Water in the Southern Ocean. *Journal of Physical Oceanography*, *51*(9), 2951–2972.

---

## Referee Comment (RC2)

**Review of "Southern Ocean deep mixing band emerges from a competition between winter buoyancy loss and upper stratification strength" by R. Caneill, F. Roquet and J. Nycander, submitted to EGUsphere.**

**Overview:** This paper examines the deep mixing band (DMB) in the SubAntarctic Zone of the Southern Ocean. It presents a useful (& novel in this context) diagnostic of the DMB based on wintertime surface buoyancy loss and pre-winter ocean stratification. To the best of my knowledge this is possibly the first such diagnostic to identify the geographical location of the DMB.

It then proceeds to analyze the impact of having temperature dependence of the thermal expansion coefficients. I think this part is a bit over-stated, but interesting.

Overall I think this is a well-written paper with robust findings, of interest to EGU journals such as Annales Geophysicae, Ocean Science etc.

I recommend a minor revision with comments below.

Signed, Justin Small, NCAR

**Minor comments:**

Lines 17-20, Fig. 1. The text mentions Orsi et asl. While the figure quotes Park et al. for identifying fronts. Which do you use?

Lines 33. I was surprised that the following important papers on SAMW were not referenced, as they are relevant to this paper:
Cerovečki I, Talley L D, Mazloff M R, Maze G (2013) Subantarctic mode water formation, destruction and export in the eddy-permitting Southern Ocean state estimate. J Phys Oceanogr 43: 1485-1511
Cerovečki, I., and M. R. Mazloff, 2016. The spatiotemporal structure of processes governing the evolution of Subantarctic mode water in the Southern Ocean. J. Phys. Oceanogr., 46, 683-710.
These papers give an alternative viewpoint on where the DMB and SAMW forms from Watermass Transformation theory.

Line 36. I wonder if the following references are more related to the DMB:
Frölicher TL, Sarmiento JL, Paynter DJ, Dunne JP, Krasting JP, Winton M (2015) Dominance of the Southern Ocean in Anthropogenic Carbon and heat uptake in CMIP5 models. J Clim 28:862–886
Li et al. 2023 https://www.nature.com/articles/s41467-023-42468-z

Lines 96-109. As an aside, CORE (Large and Yeager 2009) and JRA55-do (Tsujino et al. 2018) follow a similar approach of globally-adjusting surface fluxes. You may also want to view Fig. 17 of Small et al. (2021) which compares 4 products for surface heat flux. There the OAFLUX product was 0.25deg., personally provided by Lisan Yu, which will hopefully be freely available soon (https://oaflux.whoi.edu/data-1-4o/).

Lines 150-152. The stratification measure shown in Eq. 9 was used by Lee et al. 2011 and Small et al. 2021 who should be referenced.
Lee M-M, Nurser AJG, Stevens I, Sallée J-B (2011) Subduction over the Southern Indian Ocean in a high-resolution atmosphere-ocean coupled model. J Clim 24:3830–3849

Line 185. There is also large buoyancy loss off eastern Australia, an eastern boundary current!

Re discussion of Fig. 4: the temperature and salinity parts of B_EK strongly compensate (i.e. SST gradients at fronts are compensated by SSS gradients) so that their net effect (Fig. 4c) is small compared to surface heat.

Fig. 5. State that these are zonal means (presumably).

Line 225-226. The sign conventions are confusing. The temperature component of stratification is negative but B250 is positive (in near-polar regions). Maybe just use one metric, B250, and be clear of the meaning of the sign.

Line 235. I would remove the Small et al. citation from this line, replace with DuVivier et al. 2018, also Yeager and Large 2007: https://doi.org/10.1175/2007JPO3629.1

Line 236. Begin this paragraph with a statement like "We now consider the total stratification, $B\_S\_250+B\_theta\_250…$"

Fig. 7 caption. Panel a) and b) titles are swapped.

Fig. 7a, line 239. You could relate this to Small et al. 2021 their Fig. 3 and Supp. Fig. S4, which confirms that the DMB has weak stratification year-round – i.e. preconditioning.

Fig. 8 is a nice figure but is not sufficiently described in the text.

Lines 243-251. This discussion of Fig.5 can be moved earlier, before the discussion of Figs. 6,7.

Section 4. The effect of non-constant TEC is interesting, but I do not think it is a first order factor for the DMB. The DMB will have overall similar characteristics with constant TEC (Fig. 11c) – although it is likely to be deeper and wider in the Pacific, and the Indian Ocean will have high latitude deep mixed layers. You say this in lines 298 -304. Perhaps you can slightly de-emphasize the importance of TEC.

Lines 324-328 seem to discuss a small effect. Perhaps delete this paragraph.

Line 345. Finish the first sentence of this paragraph with "using constant TEC."

Lines 331-33 and 353-355. I think an interesting follow-on question is why the stratification in SE Pacific is so weak year-round? (see also Fig. 3a of Small et al. 2021). Sloyan et al. 2010 is a

good reference which I was not aware of before. Perhaps it is relevant here that in the South-East Pacific the atmosphere storm track, and associated waves, are present year-round? You could also look at summertime air-sea buoyancy flux, and whether it receives much buoyancy gain in summer.

Finally, useful relevant papers are Qiu and Chen 2006 and Yu et al. 2020
Qiu, B., and S. Chen, 2006: Decadal variability in the formation of the North Pacific subtropical mode water: Oceanic versus atmospheric control. *J. Phys. Oceanogr.*, 36, 1365–1380, https://doi.org/10.1175/JPO2918.1.
Yu et al. 2020   https://doi.org/10.1175/JCLI-D-20-0119.1

These papers look at interannual variability of mixed layer depth, and how it depends on pre-winter stratification vs surface fluxes.

For future work (this is not a review request, just a comment), it could be interesting to look at interannual variability in high-resolution models, including HighResMIP, also Small et al. (2014) https://www.earthsystemgrid.org/dataset/ucar.cgd.asd.output.html
And Chang et al. 2020 https://doi.org/10.1029/2020MS002298

---

## Author Comment (AC1)

**Answer to Qian Li,**
**Southern Ocean deep mixing band emerges from a competition between winter buoyancy loss and upper stratification strength**

Romain Caneill, Fabien Roquet, and Jonas Nycander

December 2023

Dear Qian Li,

Thank you for your careful consideration of our manuscript. We believe that based on your comments, we were able to improve the clarity of our manuscript.

Please find our answers to the reviews in this document. We used `this monospace font` to cite the original comments, and we provided point-by-point responses.

Best regards,

Romain Caneill and co-authors

**Major comments**

1. `Lines 305-307`

    (a) `What is the definition of ''`**`narrowness`**`'' of DMB here? What is the horizontal resolution of observational data and ECCO outputs used in this study? Compared with the MLD simulated from an eddying (0.1°) ocean model (e.g., Fig. 2 in Li & Lee, 2017), the meridional extent of wintertime MLD (Fig. 1b) looks still quite broad to me. Note that the MLD from de Boyer Montégut et al. (2023) is at` $1° \times 1°$ `spatial resolution. More discussion/clarification is needed here.`

    Thanks for the question. Narrow can indeed have different meanings depending on the context. Here it should be understood as "having a belt shape, i.e. being more elongated than wide, and representing only a small region (in latitude) of the Southern Ocean". We replaced "narrowness" by "limited latitudinal extent" to clarify.

    The observation data have 1 degree of resolution for the fluxes, and MIMOC (used for columnar buoyancy) is at $0.5° \times 0.5°$ resolution. We added a sentence in methods (Sect. 2) providing the resolution.

    Eye inspection of Fig. 2 in Li & Lee, (2017) gives a latitudinal extent of 5 to 10 degrees of the 250 m contour, which is consistent with the white contour defining the DMB (Fig. 1 (a) of our manuscript). We use a different colormap saturating at 350 m, instead of 550 m in Li & Lee (2017). We belive that this is the main reason the DMB appears "narrower" in the latter figure.

    (b) `The authors also need to clarify that ''this balance is sufficient...'' on the timescale of annual mean or 6-month-mean, which is the time period focused in this study. In fact, I am concerned on comparing the contributions of surface buoyancy fluxes and Ekman buoyancy transport on the timescale of 6-month or longer, as they both can play the dominate role of preconditioning in the deep MLD formation but during different periods. For example, the May net air-sea heat flux and June Ekman heat advection are both critical in the August/September MLD formation (Li & England, 2020).`

    This is true that this study focuses on large spatial scale, annual (or 6-month) state, and climatological state. We added to the sentence that the balance is sufficient to predict the DMB in the climatological state.

    Regarding your question about comparing the contribution of surface buoyancy fluxes and Ekman transport of buoyancy: both are considered on a time-mean basis during the cooling season, which

spans 6 months. In our analysis, we account for both of their individual effects, integrated along all the cooling season. We added a sentence in section 3.2 mentioning that the Ekman transport can participate in the interannual variability of the MLD (Cerovečki et al., 2019; Li & England, 2020).

(c) The first-order role of surface buoyancy fluxes is mostly considered on a large-scale or a zonal average (Fig. 5). In southeast Pacific, Ekman buoyancy transport actually dominates the deep MLD formation and Subantarctic Mode Water (SAMW) formation (Cerovečki et al., 2019; Li & England, 2020). More discussion is needed here.

Thanks for this comment. The balance done in the study is between the stratification, and the *total* buoyancy fluxes (surface + Ekman). Ekman transport is thus taken into account. We realised that it was not entirely clear in the manuscript, as $\mathcal{B}^{CS}$ was defined in Sect. 2.1.1, before we introduce the Ekman buoyancy transport. We clarified the manuscript by moving the definition of $\mathcal{B}^{CS}$, and explicitly writing that it is the sum of the surface fluxes and Ekman transport. Moreover, we did not aim to compute a full MLD budget, but only compare the balance between two important quantities (stratification and buoyancy loss). Computing the balance between the stratification and the *surface* buoyancy loss during the cooling season (i.e. not taking the Ekman transport into account) predicts the DMB with a quite good accuracy (Fig. 1 of this document). We thus do not find that Ekman transport is of primary importance in setting the DMB. It still acts as a secondary process that make the DMB slightly more larger.

(d) In the MLD budget, there are some other terms, such as vertical Ekman pumping and vertical mixing, that could be potentially important. More discussion is needed here.

We already mentioned in the conclusions that mixing contributes to deepening the mixed layer, we added a sentence to mention the Ekman pumping. As the balance between the buoyancy loss and the stratification is sufficient to predict the deep mixing band, other processes will likely modify slightly the existing deep mixing band, but do not think that they may dominate as their spatial distribution does not match the DMB.

2. Section 2.2: I do not follow why the authors use a **deep mixed layer threshold of 250 m** in the estimation of stratification intensity. The wintertime MLD, that forms north of the Subantarctic Front (SAF), can be much deep over 500 m. The summertime MLD at southern high-latitudes can be less than 100 m. Thus, it is unjustified to apply this threshold across the entire Southern Ocean. I suggest to use the actual MLD in the calculations (Eq. 11). Then, this equation can be written as follows:

$$B_{\mathrm{MLD}} = \frac{g}{\Delta t} \left( \int_{-\mathrm{MLD}}^{0} \alpha(z) \frac{\partial \theta}{\partial z} z dz - \int_{-\mathrm{MLD}}^{0} \beta(z) \frac{\partial S}{\partial z} z dz \right)$$

We think that this comment comes from a confusion between the mixed layer depth and its threshold used to define the deep mixing band (250 m), and the depth at which we compute the columnar buoyancy. As the focus of this study in on the deep mixing band, we must use the same depth everywhere for the columnar buoyancy, otherwise the balance between $B_{MLD}$ and $\mathcal{B}^{CS}$ would not predict the DMB. It could be a useful quantity to verify globally where the mixed layer is produced by buoyancy loss, and where other processes are playing a dominant role. We leave this question for another study.

We clarified the text regarding this, and we added a figure in the manuscript's appendix (reproduced in Fig. 2 of this document) to show that our result do not depend on the exact value of this depth, as long as the depth is the same to define the deep mixing band and for computing the columnar buoyancy.

**Minor comments**

- Figure 8: The authors examine three different transects in the Atlantic, Indian, and Pacific sectors of the Southern Ocean. However, these transects cover a large domain, in which many different processes may mix together. I recommend changing each domain to that more localized in the deep MLD formation region. For example, the SAMW formation regions analyzed in Li et al. (2021) and Cerovečki et al. (2013).

[Figure]

Figure 1: Same as Fig. 7 of the manuscript, but using only the surface component of the buoyancy fluxes.

[Figure]

Figure 2: Comparison between different thresholds for the definition of the deep mixing band, and for computing the columnar buoyancy (added in Appendix of the manuscript).

In Cerovečki et al. (2013), they define the sectors with longitude bounds "150°E–70°W for the Pacific sector, 70°W–20°E for the Atlantic sector, and 20°–150°E for the Indian sector". Our 3 transects are taken in the middle of the three sectors (transects in the middle of the Atlantic sector at 330°E, in the middle of the Indian sector at 100°E, and in the middle of the Pacific sector at 210°E). The three transects we provide are thus already localised (they are not means for each sector).

- Equation (3): Define the $\tau^x$ and $\tau^y$.

  Thanks for seeing that the definition was missing. We added it.

- Equation (4)-(7): Define the $\theta$ and $S$.

  Thanks for seeing that the definition was missing. We added it.

- Equation (8): Define the $Z$ and $z$ right after this equation.

  We added the definitions.

- Figure 6: Add ''annual'' to the figure caption.

  We added to the caption that the stratification is computed in April (it is not an annual stratification).

- Line 331: Change Fig. 3 (f) to Fig. 4 (f).

  Thanks for catching this typo, we corrected.

- There are too many acronyms in the paper, and I suggest to reduce the use of them if possible. For example, I may suggest to spell out the ''Southern Ocean (SO)'', ''cooling season (CS)'', ''warming season (WS)'', etc.

  We believe that most of the acronyms we use are commonly used (e.g. SO, the name of the fronts, the ACC). We acknowledge that CS and WS are new to this study (to our knowledge), without bringing much new information. Except in equations, we replaced "WS" with "warming season". Except in equations, we replaced "CS" by "during the cooling season" or an equivalent formulation.

---

## Author Comment (AC2)

**Answer to Justin Small,**
**Southern Ocean deep mixing band emerges from a competition between winter buoyancy loss and upper stratification strength**

Romain Caneill, Fabien Roquet, and Jonas Nycander

December 2023

Dear Justin Small,

We appreciate your very relevant comments on our manuscript, and we believe that your suggestions will improve its clarity and scientific value.

Please find our answers to the reviews in this document. We used `this monospace font` to cite the original comments, and we provided point-by-point responses.

Best regards,

Romain Caneill and co-authors

- `Lines 17-20, Fig. 1. The text mentions Orsi et al. While the figure quotes Park et al. for identifying fronts. Which do you use?`

  We use the Park et al. climatology of the fronts (as written in the caption of Fig. 1). We added the Park et al. citation in the text.

- `Lines 33. I was surprised that the following important papers on SAMW were not referenced, as they are relevant to this paper: Cerovečki et al (2013), and Cerovečki and Mazloff (2016). These papers give an alternative viewpoint on where the DMB and SAMW forms from Watermass Transformation theory.`

  Thank you for pointing to these two references. They are indeed important, we added them.

- `Line 36. I wonder if the following references are more related to the DMB: Frölicher et al. (2015) and Li et al. (2023)`

  Thanks for these two references. It is true that the two references we cite at Line 36 are more general, but still relevant. We added your suggestion to the manuscript.

- `Lines 96-109. As an aside, CORE (Large and Yeager 2009) and JRA55-do (Tsujino et al. 2018) follow a similar approach of globally-adjusting surface fluxes. You may also want to view Fig. 17 of Small et al. (2021) which compares 4 products for surface heat flux. There the OAFLUX product was 0.25deg., personally provided by Lisan Yu, which will hopefully be freely available soon (https://oaflux.whoi.edu/data-1-4o/).`

  It is true that closing the heat budget is a hard task.... The new 0.25deg OAFlux product seems to be a major improvement in closing the budget, but it is not available yet.

- `Lines 150-152. The stratification measure shown in Eq. 9 was used by Lee et al. 2011 and Small et al. 2021 who should be referenced.`

  Thanks for pointing it out, we added the references.

- `Line 185. There is also large buoyancy loss off eastern Australia, an eastern boundary current!`

  You are right, the Leeuwin Current flows southward and induces large heat loss to the atmosphere. We added it to the sentence.

- Re discussion of Fig. 4: the temperature and salinity parts of $B^{Ek}$ strongly compensate (i.e. SST gradients at fronts are compensated by SSS gradients) so that their net effect (Fig. 4c) is small compared to surface heat.

  Thanks for pointing that. It was not explicitly mentioned in the text, we added it.

- Fig. 5. State that these are zonal means (presumably).

  The solid lines are the zonal medians. We added this in the caption.

- Line 225-226. The sign conventions are confusing. The temperature component of stratification is negative but B250 is positive (in near-polar regions). Maybe just use one metric, B250, and be clear of the meaning of the sign.

  The confusion may have arisen from the swap in the caption of Fig. 7. B250 is negative everywhere, even in the polar region (Fig. 7a). We explicitly added that $B_{250}$, $B_{250}^{\Theta}$, and $B_{250}^{S}$ are negative under stabilising conditions.

- Line 235. I would remove the Small et al. citation from this line, replace with DuVivier et al. 2018, also Yeager and Large 2007: https://doi.org/10.1175/2007JPO3629.1

  Thanks for pointing it out, we changed the references.

- Line 236. Begin this paragraph with a statement like ''We now consider the total stratification, B_S_250+B_theta_250...''

  Thanks for the clarification, we updated the text.

- Fig. 7 caption. Panel a) and b) titles are swapped.

  It is the caption that had the swap. We corrected.

- Fig. 7a, line 239. You could relate this to Small et al. 2021 their Fig. 3 and Supp. Fig. S4, which confirms that the DMB has weak stratification year-round -- i.e. preconditioning.

  Thanks for pointing to this figure of the Small et al. 2021 paper, which shows similar pattern. We added the reference and the remark on preconditioning.

- Fig. 8 is a nice figure but is not sufficiently described in the text.

  Thanks for your appreciation. We added a few sentences to describe better Fig. 8.

- Lines 243-251. This discussion of Fig.5 can be moved earlier, before the discussion of Figs. 6,7.

  We moved the discussion of Fig. 5 to the beginning of section 3.3.

- Section 4. The effect of non-constant TEC is interesting, but I do not think it is a first order factor for the DMB. The DMB will have overall similar characteristics with constant TEC (Fig. 11c) { although it is likely to be deeper and wider in the Pacific, and the Indian Ocean will have high latitude deep mixed layers. You say this in lines 298-304. Perhaps you can slightly de-emphasize the importance of TEC.

  We agree that the TEC variations do cause the formation of the DMB, however, they impact its location and *narrowness*. It is true that in the Indian Ocean, with a constant TEC, the predicted DMB has a similar shape as the observed one. This is not true in the Pacific Ocean, where the predicted DMB extends to the sea-ice edge. In this sense, we respectfully disagree with the reviewer and consider that variations in the TEC are first-order factor controlling the DMB.

  We slightly modified the last sentence of the paragraph to emphasise that the southern boundary of the DMB is strongly constrained by the TEC in the Pacific Ocean, while in the Indian Ocean the TEC variations prevent the formation of a second deep ML region south of the DMB.

- Lines 324-328 seem to discuss a small effect. Perhaps delete this paragraph.

  We shortened and moved this paragraph into the methods (Section 2.1.2).

- Line 345. Finish the first sentence of this paragraph with ''using constant TEC.''

Thanks for clarifying this point, as only using a variable TEC can lead to a correct comparison between heat and freshwater fluxes. We corrected the sentence.

- Lines 331-33 and 353-355. I think an interesting follow-on question is why the stratification in SE Pacific is so weak year-round? (see also Fig. 3a of Small et al. 2021). Sloyan et al. 2010 is a good reference which I was not aware of before. Perhaps it is relevant here that in the South-East Pacific the atmosphere storm track, and associated waves, are present year-round? You could also look at summertime air-sea buoyancy flux, and whether it receives much buoyancy gain in summer.

Thanks for this interesting question. We can see on Fig. 4 (f) that the SE Pacific is a region of negative annual buoyancy fluxes. It is also a region with only small buoyancy loss during the cooling season (Fig. 7 (b)). The summer buoyancy fluxes are positive, but quite small in this region. Precisely looking at the processes leading to the only small summer buoyancy gain would be an interesting follow-up study.

- Finally, useful relevant papers are Qiu and Chen (2006) and Yu et al. (2020). These papers look at interannual variability of mixed layer depth, and how it depends on pre- winter stratification vs surface fluxes.

Thanks for these two interesting papers that we were not aware of. They could be the basis of a future work assessing variations in the position and properties of the DMB.

---

## Referee Report (RR1)

**Second Review: "Southern Ocean deep mixing band emerges from a competition between winter buoyancy loss and upper stratification strength"**

Authors: Romain Caneill, Fabien Roquet, and Jonas Nycander

**Summary**

The authors did a nice work in response to my previous review. I think the manuscript is significantly improved. I have no major issues with their main conclusions. I have however, a few minor comments regarding the work done in this revision, listed below.

**Minor comments**

– Lines 178–179: "*We expect $B_{250}^{\Theta}$ to be positive north of the PF (stabilising effect of temperature) and negative south of it (destabilising effect), while $B_{250}^{S}$ should be negative north of the SAF and positive south of it.*" I think "positive" and "negative" should be swapped here.

– Figure 7 and line 5: What time period is defined for "late summer"? In Fig. 7, why not compute the terms of $B_{250}$ and $B^{CS}$ during the same time period, such as the cooling season? Otherwise, it needs to justify the meaning/fairness to compute the difference between them $(B_{250} - B^{CS})$.

---

## Author Response (AR2)

**Answer to Qian Li,**
**Southern Ocean deep mixing band emerges from a competition between winter buoyancy loss and upper stratification strength**

Romain Caneill, Fabien Roquet, and Jonas Nycander

February 2024

Dear Qian Li,

Thank you for your careful second consideration of our manuscript. We improved the points you mentioned. Please find our answers to the reviews in this document. We used `this monospace font` to cite the original comments, and we provided point-by-point responses.

Best regards,

Romain Caneill and co-authors

**Minor comments**

- `Lines 178 -- 179: ''We expect` $B_{250}^{\theta}$ `to be positive north of the PF (stabilising effect of temperature) and negative south of it (destabilising effect), while` $B_{250}^{S}$ `should be negative north of the SAF and positive south of it.'' I think ''positive'' and ''negative'' should be swapped here.`

  Thanks for catching up this typo. As you mentioned, as temperature is increasing stratification north of the PF, this means that $B_{250}^{\theta}$ is negative (similar reasoning applies for salinity). We corrected it in the manuscript.

- `Figure 7 and line 5: What time period is defined for ''late summer''? In Fig. 7, why not compute the terms of` $B_{250}$ `and` $\mathcal{B}^{CS}$ `during the same time period, such as the cooling season? Otherwise, it needs to justify the meaning/fairness to compute the difference between them` $(B_{250} - \mathcal{B}^{CS})$.

  Here, late summer is defined as the time before which the ocean starts loosing buoyancy, which is in April on average. We do not compute the average $B_{250}$ during the cooling season, as we use the columnar buoyancy as the starting point, from which buoyancy is removed during the cooling season. Thus, taking $(B_{250} - \mathcal{B}^{CS})$ provides the state of the columnar buoyancy at the end of winter, neglecting other processes than surface buoyancy fluxes, and Ekman driven buoyancy advection.

  The explanation given line 168 (section 2.2) was clarifying this idea: "Thus the comparison of $B_{250}$ with the buoyancy loss directly informs if the buoyancy fluxes can produce a mixed layer of $250\,\text{m}$."

  We added a comment to better clarify: "We compute $B_{250}$ in April, just before the cooling season, so $(B_{250} - \mathcal{B}^{CS})$ estimates the columnar buoyancy at the end of winter. Thus, the comparison of $B_{250}$ with the buoyancy loss directly informs if the buoyancy fluxes can produce a mixed layer of $250\,\text{m}$."